# The Cost of a Sustainable Water Supply at Network Kiosks in Peri-Urban Blantyre, Malawi

**Andrea B. Coulson** [1,*], **Michael O. Rivett** [2,3], **Robert M. Kalin** [2], **Sergio M. P. Fernández** [2,4], **Jonathan P. Truslove** [2,5], **Muthi Nhlema** [6] and **Joseph Maygoya** [7]

1   Department of Accounting and Finance, Strathclyde Business School, University of Strathclyde, Glasgow G4 0QU, UK
2   Department of Civil and Environmental Engineering, University of Strathclyde, Glasgow G1 1XN, UK; rivett@groundh2oplus.co.uk (M.O.R.); robert.kalin@strath.ac.uk (R.M.K.); sergio.fernandez@wfp.org (S.M.P.F.); jonathan.truslove@ewb-uk.org (J.P.T.)
3   GroundH$_2$O Plus Ltd., Quinton, Birmingham B32 1DY, UK
4   Engineering Unit, World Food Programme, 653 A Block 68, Arkawit, Khartoum P.O. Box 913, Sudan
5   Engineers Without Borders, London SE11 5RR, UK
6   BASEFlow, Galaxy House, Chichiri, Blantyre P.O. Box 30467, Malawi; muthi@baseflowmw.com
7   Water for People, Blantyre P.O. Box 1207, Malawi; jmagoya@waterforpeople.org
*   Correspondence: a.b.coulson@strath.ac.uk

**Abstract:** Empirical insights were made into the challenges of supplying water to communities within low-income areas of peri-urban Blantyre, Malawi. A networked public water supply is provided to those without a domestic tap via communal water kiosks managed by community-based Water User Associations (WUAs) under a government mandate. There has been considerable debate surrounding the tariff charged for water supplied to such vulnerable communities. However, research has largely failed to consider the costs of WUAs operating the kiosks and the impact on the kiosk tariff. The determination of kiosk tariffs is critical to ensuring lifeline access to a sustainable water supply under Sustainable Development Goal 6. We provide evidence of this from our experience in the field in Blantyre. In particular, we argue that sustainable kiosk running costs cannot be born solely by the end user. A number of reforms are needed to help reduce the kiosk tariff. To reduce WUA costs and the kiosk tariffs, WUAs need more training in financial record keeping and cost management, WUAs should not inherit outstanding kiosk debt upon taking over their operations, and water boards should build kiosk costs over which they have fiscal responsibility into integrated block tariff calculations and subsidize them accordingly.

**Keywords:** kiosk; tariff; water supply; Water Users Associations (WUAs); affordability; accounting; service delivery; peri-urban; Sustainable Development Goal 6

## 1. Introduction

Achieving a sustainable drinking water supply for the poor represents an on-going challenge globally. Between 2000 and 2015, this ambition was embedded in the Millennium Development Goals (MDGs), and remains enshrined within Sustainable Development Goal (SDG) 6, to "ensure availability and sustainable management of water and sanitation for all" by 2030. Challenges relate to both the maintenance of water availability and how to manage the holistic system of water supply. These span the mode of water distribution and the supply chain reaching and interfacing with the user community. It also includes water pricing and tariff structures, and the affordability and economics of supply, including socio-economic influences, which are, together, overarched by sustainable governance, management systems and accounting employed.

'Free' supply of water historically to those living in poverty via a standpipe has faced challenges to ensure the long-term sustainability of this provision [1]. If the supply of

potable water is free to those living in water poverty, the cost of water supply and maintenance of the supply service would need to be financed indirectly from other sources, such as subsidy by local, national and international governments or aid providers. Particularly in the developing world, meeting the costs of operation and the maintenance of a networked water supply to everyone are considerably challenged by limited public budgets, lack of economic development and lack of political will. This challenge is accentuated in peri-urban settings where water is often provided via an expensive public network. Typically operated by a large water utility/board, this brings dependencies not found in rural water supply settings where communities may manage their own supply independent of other parties, typically hand-pumped groundwater boreholes serving several hundred people [2]. Access to and affordability of water kiosk tariffs is of importance to public policy makers from national and district government, funders including international and local NGOs and, most critically, water users, people living in water poverty purchasing this water supply service.

Given financial pressures and the criticality of demand, the 1970–1980s' widespread practice of providing free potable water via standpipes to the poor has increasingly been replaced by water kiosks charging a tariff to supply water at standpipes [3–7]. The tariff is intended to cover the costs of the water supply service; however, those unable to pay may be reduced to drinking water unfit for consumption.

While there has been considerable debate on the pros and cons of tariffs charged for water supply across the developing world [6,8–12], kiosk tariffs have received little attention. They have been recognized within marginal costing methods, such as the Integrated Block Tariff (IBT), as the lowest tariff limited to the cost of delivering water supply. Emphasizing the importance of water to people living in poverty, this has also been referred to as being a "lifeline block" tariff by NGOs like WaterAid [4].

The method of determining the tariff is critical to ensuring affordability, access to water and the maintenance of supply, all important elements of sustainability. Who should bear the costs of providing the water supply becomes a question fundamental to that sustainability. For example, Tchuwa [13] drew attention to historical inequalities in access to water resulting from a common charge to all domestic users, affluent or poor, to support the costs of supply. Further, a recent study by advocates of sustainability called for multi-criteria approaches to decision-making, from financial cost recovery to involving local communities in service delivery to ensure access to and affordability of supply under sustainability agendas and SDG 6 [14,15].

Reviewing water governance, Pihljak et al. [16] recognize the influence of management reform apparent in water supply over recent decades, and the emergence of Water Users Associations (WUAs). WUAs have been established as participatory delegated management groups involving local communities in water supply and decision-making, aiming to make them affordable and sustainable [4,15]. Drawing case evidence from a study of Lilongwe, Malawi, Pihljak et al. [16] present evidence that WUAs were able to negotiate kiosk tariffs with the local public water board. However, they still find water kiosks tariffs far exceed the tariff paid by domestic users, as the cost of running the kiosk is built into the final tariff. Based on an earlier study of water supply in four areas of Blantyre, Malawi, Samson et al. [17] question the technical, financial, social/environmental and institutional sustainability of kiosks. The financial priority within their multivariate analysis is the need to establish an operations and maintenance fund to support overall the sustainability of supply, with the strongest support for input from a user committee to ensure the reliability of water supply. Neither of these studies on water kiosks detail how water kiosk tariffs are calculated.

WUA development around water kiosks added a layer of governance, accountability and costs to tariff calculation. Emergent case research on WUAs, principally on water management in rural agricultural settings, has emerged, which is often critical of the limited participation by locals [18–23]. However, there remains a gap in the literature on how kiosk tariffs are determined beyond a theoretical breakeven principle of cost recovery

built into an IBT's lifeline block. The costs of operating kiosks through WUAs and the relationship of these costs to the kiosk tariff has not been examined to our knowledge.

This paper examines the specific challenge of maintaining the sustainability of water supply to people living in water poverty, and aims to shed more light on the costs of operating network fed water kiosks in peri-urban Blantyre, Malawi. Our research questions are: How are water kiosk tariffs calculated? How does this calculation relate to the WUA's costs of operating the kiosk? We aim to provide insight into who is, and who should be, paying for these costs. We contribute to the emergent literature on kiosk tariffs by providing empirical insight into the calculation of kiosk tariffs and how they relate to the costs of WUAs managing kiosks. We base our empirical study on the initial formation and development of WUAs to support Blantyre's network-fed water kiosks and their subsequent management period from 2012 to 2016 with a current update added at the time of writing. This formation at the end of the MDG period allows us to provide learning outcomes to inform policy-making under the Malawi Water Resources Act 2013 and SDGs, in particular SDG6.

Building on Pihljak et al.'s [16] findings in Lilongwe, we provide similar evidence of a two-tier tariff system incorporating the IBT plus an increment to cover the WUA running costs in Malawi. The calculation methods of the tariff by Blantyre Water Board (BWB), who retain fiscal responsibility for water supplies, are provided in detail, and evidence of WUAs costs, mandated to manage kiosk delivery on a participatory not-for-profit basis, is examined. We provide evidence of consumers bearing the costs of kiosk delivery and question whether this is affordable, as it exceeds domestic rates and, thus, accessible as intended under SDG6. Like Samson et al. [17], we cast doubt on the sustainability of the water supply in Blantyre and we highlight the need to change tariff setting and local WUAs to achieve progress towards SDG6. Our findings offer insight into how to make water supply at the kiosk more equitable and sustainable. In particular, as there is no statutory guidance on legacy debt, we propose future reforms to the Water Works Act (Malawi) and updates to the National Water Policy should take this into account.

## 2. Materials and Methods

Key technical context and the case setting are detailed below, followed by a description of the research methods employed in our longitudinal action research study of peri-urban Blantyre, Malawi.

### 2.1. Technical Setting: The Theory and Practice of Kiosk Tariffs

The core water tariff objectives defined by Laredo [7] are: financial sustainability and cost recovery; efficient allocation of scarce sector resources; income distribution and fiscal viability. When water is supplied via a kiosk, tariffs based on purely financial calculations are insufficient to make adequate consideration of socio-economic conditions and the vulnerability of people to the set tariffs. Financial information needs to be supplemented with information from the community, cognizant of the social impact of clean water supply on the poor (see, for example, OECD [12]). The ultimate aim is improved access to clean water for all.

#### Economic Framing

As noted in the introduction, a popular form of tariff in developing countries including Malawi is the Integrated Block Tariff (IBT) for water networks. The tariff is a charge for the provision of a networked water delivery service rather than a payment for the water itself, covering the cost-of-service provision including, for example, operations and maintenance of a piped network. The block tariffs are established based on an assumed ability-to-pay basis and the volume of water usage per billing period is allocated into a number of discrete blocks. Each block builds upon the previous one to form an incremental pricing system in which thresholds are applied to water usage, after which additional prices come into force for the service provision. Domestic users are assumed to occupy the lowest tariff

block, most commonly moving up to industrial users, who require the highest volume of water. As noted previously, inequalities can occur when all domestic users, irrespective of ability to pay, are charged the same tariff. Further, the introduction of WUAs operating water kiosks has introduced a second service modality beyond direct network supply. Alternatively, WUAs are also required to operate kiosks in rural settings fed, for example, from boreholes or gravity-fed (piped) supply systems. A separate body of research outside the scope of this paper is relevant in rural settings [2,24].

In terms of IBT systems, the costs of water supply connection to the network may be borne by the particular user group for which they are established, and government subsidy may be possible to address inequalities. Therefore, an industry, public service organization or domestic user will pay to have their own 'tap' fitted and are then charged a block tariff for water usage based on the quantity supplied. Within an IBT system, it is assumed that connection costs will also be proportional to the tariff charge, with the highest costs for industry and the lowest costs for lifeline users. Research has recognized that, for the poorest domestic user, the tariff should be no more than the breakeven cost of service provision [4]. However, if we factor in the connection costs that need to be borne by the respective users, the tariff could change considerably. In situations where WUAs are established to operate the kiosks and sell the water provision service, the costs could be higher than for domestic users and become prohibitive to many. This paper explores the following questions: How are water kiosk tariffs calculated? How does this calculation relate to the WUA's costs of operating the kiosk? The kiosk connection cost and WUA operating costs need to keep ability to pay in mind.

### 2.2. Case Setting

Taking our empirical study into account, Malawi is a low-income country ranking 172 out of 189 in the 2019 Human Development Index, with a population of over 17.5 million that is growing rapidly at up to 3% per annum [25]. In Malawi's commercial city of Blantyre, the sustained delivery of safe drinking water remains an on-going challenge. Public water supply is provided by the public utility, Blantyre Water Board (BWB), and is mostly drawn from the Shire River, the sole natural outflow from Lake Malawi. Towards 100 Ml/d (mega-litres per day) is pumped uphill through a 48-km pipeline to the city overcoming an elevation of 800 m, with additional booster stations necessary to distribute water throughout the hilly city terrain, leading to 40% of operating costs being consumed by electricity [26]. BWB charges a tariff for all water supplied from this source at the kiosks and there is evidence the water supply is vulnerable to common network failures. This is not surprising given the above logistical challenges, which are compounded by the intermittency of electricity supply, decreased water availability during dry season, and major drought and flood events. This is before we take account of the socio-economic and political influences of power which interrupt tariff-setting principles, policy objectives and access to the water supply [13,16,26–32].

Limited financial capacity causes government dependency on infrastructure investment from international partners, such as the World Bank, European Investment Bank and NGOs, to build capacity and maintain supplies [27]. Tchuwa [13] demonstrated how this has influenced the state ownership and control of Malawi and, in particular, Blantyre's water infrastructure. This led, between 1980 and 2000, to reductions in subsidy for low-income users and the protection of tariffs paid by the poor. Historically, water kiosks supplying low-income areas were managed by a variety of operators, including BWB, but also community, religious and political institutions, leading to further problems with access to water by the poor. An analysis of everyday life in Malawi by Cammack [33] reveals, at the extreme, evidence of political organizations capturing kiosk money for party funds and restricting water distribution based on party membership. Such actions meant bills for water supply were, in turn, unpaid and supplies then disconnected, a particular problem at the turn of the millennium [26].

Widespread WUA introduction to govern kiosks were considered critical to providing and maintaining water-supply to low-income areas (LIAs). In September 2007, Blantyre City Council, Blantyre Water Board and the NGO Water for People signed a Memorandum of Understanding to establish formally defined WUAs responsible for the management of kiosks [34]. In 2010, new "Guidelines for the establishment of water users association in Malawi" were published [35]. Each WUA was to be formed as a not-for-profit legal entity composed of a General Assembly (GA), Executive Board and a secretariat. Maoulidi [26] provides evidence of a growing number of WUAs taking over operations, maintenance and revenue collection for kiosks. There is little research to date of WUAs in Blantyre.

WUA form and operating procedures also became explicit within Malawi's Water Resources Act 2013 [36]. The participative management philosophy embedded in WUA establishment is important and requires its own body of research to determine its success. A review of governance is outside the scope of this paper. We are focused on the cost implications of the requirement for WUAs and their particular form.

A recent study by Pihljak et al. [16] used an everyday lens to examine the disparate water tariffs in Lilongwe, Malawi. Taking a phased approach to the study of water tariffs between 2008 and 2018, they identified a differential approach to tariff setting based on direct water supply by the Lilongwe Water Board (LWB) and the partnership between the LWB and WUAs [16]. They provide evidence that, in Lilongwe, the tariff for direct bulk water supply to the majority of connections, including in-house supply, is set by the Government in cooperation with the LWB. However, the kiosk tariff charged to users was found to be set in a negotiation process between the LWB and the WUAs [16]. While Pihljak et al. [16] provide evidence that tariff for bulk water and the first block of in-house connections are subsidized by the government and LWB, it is not clear whether costs incurred at the kiosk by the WUA are directly subsidized. The extent to which this practice in Lilongwe is similar to the practice in Blantyre will be considered in our study. In particular, our focus on calculative practices to set tariffs and the processes in which they are embedded in everyday life pinpoints our contribution to understanding the financial costs in more detail.

Accounting for Costs at the Community Level

It is noted under the Water Resources Act 2013 that the WUAs in Malawi should not operate for profit but the costs of the service provision need to be covered. Who is responsible for all aspects of these costs is an important question. Arguably, the servicing of 'connection' costs by WUAs at water kiosks could be quite considerable and ultimately exceed those of, for example, industry or domestic users. Access to capital is an important consideration here. If a domestic user has a home environment suitable for a tapped connection and can afford a one-off payment, they may ultimately pay less for the water supply tariff than users of kiosks, as was found by Pihljak et al. [16] in Lilongwe. This highlights the importance of considering, at a minimum, a multi-disciplinary approach to account for the financial costs and benefits of kiosk connections and WUA delivery mechanisms alongside the often incommensurate social impacts.

There is significant research into calculative practices and 'accounting for sustainability' which recognizes the importance of multi-disciplinary approaches to site calculative practices (see, for example, Coulson [37], Coulson et al. [38], Frame and Cavanagh [39], Frame and Brown [40], Bartelmus [41], Bebbington et al. [42], Vardon et al. [43]). Considering the accounting praxis embedded within tariff setting opens up space to integrate 'accounts' drawn from the social relationships within decision-making and the roots of new possibilities for accounting and social accountability [44,45]. In this sense, we can build on the valuable insights of Pihljak et al. [16] and Tchuwa [13] in Malawi. If the costs of WUAs are passed onto lifeline users and become unaffordable at a local level, alternative mechanisms for community participation (reducing costs) and cost recovery (offsetting costs and subsidy) may need to be considered.

We base our empirical study on the initial formation and development of WUAs in peri-urban Blantyre to support Blantyre's network fed water kiosks. Emphasis is placed on examining the design of the tariff charged for service delivery at the kiosk and WUA accounts that demonstrate the actual costs of running the kiosk which the tariff is designed to offset.

### 2.3. Research Methods

Based on action research, we provide a longitudinal study of the tariff system in operation in peri-urban, LIAs of Blantyre City, Malawi. The study was embedded in a broader multi-disciplinary research programme conducted under the Scottish Government funded Climate Justice Fund - Water Futures Programme (CJF) (see Kalin et al. [46] and therein). The programme, led by the University of Strathclyde, aims to support the government and people of Malawi. We reflect on our experience and evidence, found while working in the field. Being part of a larger programme allowed us to build on the trusted reputation of research group members and gain access to the field through established contacts with stakeholders, supported by translation into English where necessary. Despite this, there remained linguistic and cultural challenges along with practical challenges posed by climate and local standards of living.

We have divided our findings into two main time periods. The first period, between 2012 and 2013, included working with participants from the Ministry of Irrigation and Water Development (MIWD); National Statistical Office of Malawi (NSO); Blantyre Water Board (BWB); the NGO Water for People, and eight WUAs operating in peri-urban Blantyre defined by their geographical area: Bangwe, Michiru, Mitsidi-Sanjika, Mudi, Namiyango-Chigumula, Ndirande-Matope, Nkolokoti, and Soche-Misesa. All but one of the WUAs—Nkolokoti—was the subject of our empirical research. Unfortunately, the Nkolokoti installations were victims of theft and vandalism in 2012 and all documents regarding their operation and financial data were lost.

Initially, the calculation of kiosk tariffs was explored during a series of semi-structured interviews with representatives of BWB responsible for design, implementation and account management. These were conducted at BWB premises where the BWB representatives also provided details from WUA financial accounts for which they had ultimate fiscal responsibility. Details of how accounts supported tariff calculation and revision were reviewed. All interviews, field observations and site visits were conducted by a University researcher and facilitated by representatives from NGO Water for People who had supported WUA establishment, accounting training and day-to-day kiosk operations and were an established local partner in the CJF research programme. Empirical evidence was also collected from NGO Water for People through semi-structured interviews.

Once how the tariff was calculated was established, semi-structured interviews were conducted on site with WUA secretariats and those responsible for WUA's financial bookkeeping. The location of the WUA site visit varied depending on how accounts were prepared locally and where documents were stored. Kiosk observations included interviews with water sellers recording sales and water users paying the tariff. Water for People played a critical role here in providing help with local translation. Interview evidence was triangulated with observations of day-to-day activities in the field and reviews of public and private systems documentation provided by interviewees where available. At a number of points during the research-gathering process findings were discussed and verified with BWB, the WUA secretariat and NGO Water for People. To avoid unnecessary recrimination, the authors have withheld specific quotes and copies of individual accounts from findings, as sufficient details have been provided without them to understand the accounting processes in practice and it was felt unnecessary to single out the stage of WUA development.

The second period of research took place between 2014 and 2016, coinciding with the end of the MDG era. Again, the research was facilitated by CJF Programme partner Water for People. Research involved interviews with BWB, WUAs and NGO Water for People

on changes in the practice of tariff setting. Interview findings were evaluated alongside analysis of formalized minutes of WUA meetings involving multi-stakeholder engagement. Again, interview findings and documentary reviews were triangulated with observations in the field. Minutes reviewed included, where available, General Assembly meetings at Blantyre City Council; Stakeholder Meetings at Blantyre City Council on the need for new WUAs; WUA Executive Board Meetings and AGMs; Steering Committees (every month); Task Force Meetings (every 2 weeks); and Extraordinary Board Meetings (e.g., following theft). Emphasis was placed on analyzing engagement supporting new WUAs established in Ndirande-Kachere (in 2015) and Ndirande-Malabada (in 2015). Quotes from minutes were anonymized to protect those concerned from potential recriminations.

An update on the current circumstances at the time of writing, 2021, has been added by authors still working in the field. It is noteworthy that the author associated with the NGO Water for People was not involved in stage one of the original research but helped to provide an update on current practice in the field with respect to the scope of the questions and provisional findings.

## 3. Results

### 3.1. Tariff Calculation

During the period towards the end of the MDG era, BWB operated an IBT to allocate the production costs for water supplied to a range of users. In 2013, this resulted in a charge of 2.46 Malawian Kwacha (MK) per 20 litre (L) bucket of water (Table 1), with this being the wholesale cost of bulk water supplied by BWB to the kiosk. At this time, the average US Dollar/MK exchange rate was 390.9338. This IBT tariff was then entered into what BWB refers to as the "Preliminary Bulk Meter Tariff Calculation (PBTC)" to establish, in theory, a fully costed retail rate for water at the kiosk, which allowed the kiosk seller/WUA to cover all costs of supply and break-even.

**Table 1.** IBT used by the BWB as in May 2013.

| Charging Band [1] | Charge Rate MK per L | Water Tariff (Price) | |
|---|---|---|---|
| | | 20 L Bucket | 200 L Drum |
| For water supplied for communal water points or kiosks | | | |
| Fixed rate | 0.123 | 2.46 | 24.6 |
| For water supplied for domestic purposes | | | |
| 0 to 5000 L | 0.28 | 5.60 | 56 |
| 0 to 10,000 L | 0.31 | 6.20 | 62 |
| 0 to 40,000 L | 0.36 | 7.20 | 72 |
| 0 to >40,000 L | 0.40 | 8.00 | 80 |
| For water supplied to institution | | | |
| 0 to 10,000 L | 0.52 | 10.40 | 104.00 |
| 0 to 40,000 L | 0.58 | 11.60 | 116.00 |
| 0 to >40,000 L | 0.62 | 12.40 | 124.00 |
| For water supplied for commercial purposes | | | |
| 0 to 10,000 L | 0.58 | 11.60 | 116.00 |
| 0 to 40,000 L | 0.64 | 12.80 | 128.00 |
| 0 to >40,000 L | 0.70 | 14.00 | 140.00 |
| For water supplied for industrial purposes | | | |
| 0 to 10,000 L | 0.79 | 15.80 | 158.00 |
| 0 to 40,000 L | 0.89 | 17.80 | 178.00 |
| 0 to >40,000 L | 0.99 | 19.80 | 198.00 |

[1] Only rate in uppermost charging band applies.

The PBTC included the following assumptions and standard costs based on BWB's experience of kiosk account history, recent experience of WUA kiosk management and demand considered in conjunction with a baseline survey of households conducted by Water for People [47]:

- Kiosk operates 365 days per year;

- Production cost per cubic meter for BWB is determined with reference to the IBT;
- Percentage loss of water in the BWB network = 20%, for every 1 cubic meter supplied, only 0.8 cubic meters make it to the kiosk;
- Capacity of the standard bucket for measurement at the kiosk is 20 L; water demand at kiosk = one 20 L bucket per person per day;
- Percentage of loss at the taps = 0.5% bulk water purchased;
- Maintenance = 2.5% cost of bulk water purchased;
- Overheads = 15% Sub-Total Operating Costs (STOC is cost of bulk water purchased + maintenance + salaries);
- Contingency = 5% Sub-Total Operating Costs (as above);
- Number of water sellers = 1.33 per kiosk;
- Annual salary of: administrator; office assistant; plumber/mechanic; inspector; water seller; and guard (see Table 2);
- Gross income from water sales = total volume of metered water sold at kiosks × kiosk tariff;
- Operating costs = overhead + contingency + maintenance + salaries + cost of bulk water given by IBT adjusted for 20% loss in the network and 0.5% loss at the tap;
- Net income = gross income − operating costs

**Table 2.** Adjustments made to standard costs within the PBTC May 2013.

| Standard Cost | Costs for Previous Period (MK) | Costs as at May 2013 (MK) |
|---|---|---|
| Production cost of water (IBT): | | |
| - per cubic metre (1000 L) | 80 | 123 |
| - per 20 L bucket | 1.60 | 2.46 |
| Annual salary of book-keeper becomes | 150,000 | |
| Annual cost of administrator | | 240,000 |
| Annual cost of office assistant | | 120,000 |
| Annual salary of plumber/mechanic | 150,000 | 156,000 |
| Annual salary of inspector | 150,000 | 156,000 |
| Annual salary of water seller | 72,000 | 96,000 |
| Annual salary of guard (added) | | 96,000 |

The PBTC was calculated in Microsoft Excel™ with the Tariff set at the point where net income for the WUA would equal zero and they would breakeven. When the spreadsheet was reviewed with BWB, it was found to require updating to reflect the most recent IBT and new WUA salaries. The adjustments made are given in Table 2 and illustrate increased costs and, in line with this, an increased tariff.

The kiosk tariff from the revised PBTC was 10 MK/bucket. Jobs had changed slightly, with new roles for administrators and office assistants both helping with book-keeping. Increases in standard costs were attributed to changes in economic policy and inflation. Trends in the PBTC tracked back to 2010 show tariffs increased from 3 MK to 10 MK per bucket, in line with the inflation reported by the Reserve Bank of Malawi. Of particular concern to us, we found that this tariff, paid by the community, exceeded the costs of the bulk water tariff for domestic users and was just below the cost of bulk water supplied to institutions which start at 10.4 MK. This finding supports that of Pihljak et al. [16] when studying kiosk tariffs in Lilongwe Malawi, essentially stating that kiosk tariffs were more expensive than domestic tariffs.

Most PBTC assumptions and standard costs appear reasonable, with the exception of the 20% leakage in the network charged in the PBTC. As this is beyond the control of the kiosk management, it should be attributed to the IBT. It is noteworthy that this is lower than the observed non-revenue water lost through pipe bursts and leaks, which is estimated at 50% by local NGOs. A standardized kiosk tariff was justified by BWB to

support equitable distribution. If the tariff varies between kiosks, it is reasonable to assume people living in poverty would migrate towards the cheapest supply.

### 3.2. WUA Accounts at the Kiosk

WUA employees had little influence over tariff setting. Their influence was limited to kiosk sales, cash management, book-keeping, reporting faults and recommendations to the WUA Board. BWB and Water for People helped WUAs to set up their accounting systems and trained people to use them. All sets of WUA accounts were prepared on a simple cash management basis, noting what was coming in and what was going out. At the end of the period, adjustments were made to final figures for transactions which were charged for but for which no cash had been given (i.e., money owed to BWB). To support tariff setting, BWB requested that records established by WUA include water consumption (by cubic metre) mapped against average rainfall (mm) (assuming more rainfall means more water capture for use by households, thereby reducing wet season kiosk demand), and the tariff charged per 20 L bucket. Accounting systems were found to vary from basic handwritten summary sheets recording cash to computerized spreadsheets. NGO representatives put this variance down to the formation of WUAs at different times. Early WUAs were noted as having taken part in more capacity-building interventions based on accounting and had graduated from paper-based accounting to the use of computers.

As a further reasonableness test of the PBTC, an evaluation was undertaken of actual operating costs recorded in WUA accounts. Problems with data collection occurred, as consecutive records were sometimes difficult to locate, and information was "pulled together" for our research. WUA representatives explained that, in some cases, the money transfer from water sellers to inspectors is not recorded by both parties and "money leakage could be possible". We were disappointed to find that one water seller stated: "some sellers take money for themselves, even the inspectors do". This was confirmed by a WUA Secretariat representative, noting: "... complaints about money transfers between water sellers and inspectors have occurred ... It is difficult for us to audit every single money transaction". Taken together, this evidence casts doubt on the validity of some financial statements and creates problems for income calculation and, ultimately, tariff calculation.

The operations of WUAs varied in terms of date of formation, secretariat staffing composition (standard roles and salaries applied), water sellers per kiosk and number of functioning kiosks. These characteristics are detailed in Table 3. The standard cost in the PBTC is based on 1.33 sellers per kiosk and the actual number of sellers ranged from 0.94 to 1.35 per functioning kiosk, with a standard deviation 12% of the mean. Whilst the number of WUAs is small, this variation appears relatively modest.

**Table 3.** Characteristics of seven WUAs in Blantyre as at August 2013.

| WUA | Year est. | Total Staff | Water Sellers | Other Staff | Total Kiosks | Functional Kiosks | Non-Functional Kiosks | Water Seller Per Functional Kiosk |
|---|---|---|---|---|---|---|---|---|
| Bangwe | 2012 | 33 | 26 | 7 | 43 | 20 | 23 | 1.30 |
| Michiru | 2010 | 84 | 76 | 8 | 62 | 60 | 2 | 1.27 |
| Mitsidi-Sanjika | 2010 | 89 | 77 | 12 | 66 | 65 | 1 | 1.18 |
| Mudi | 2011 | 47 | 39 | 8 | 37 | 34 | 3 | 1.15 |
| Namiyango-Ch. [1] | 2010 | 39 | 34 | 5 | 48 | 36 | 12 | 0.94 |
| Ndirande-Matope | 2009 | 68 | 58 | 10 | 43 | 43 | 0 | 1.35 |
| Soche-Misesa | 2013 | 65 | 59 | 6 | 69 | 55 | 14 | 1.07 |
| *Cost in PBTC* | | | | | | | | *1.33* |
| Total | | 425 | 369 | 56 | 368 | 313 | 55 | 1.18 |

[1] Namiyango-Chigumula; italic -it is not included in the total below.

Financial records of each WUA were reviewed from WUA formation to May 2013. We found a wide variety of bookkeeping skills demonstrated in the field, with record keeping ranging from simple pencil-written receipts and payments to detailed computerized spreadsheets. The level of costs detailed varied from simply showing total "office expenses"

set against cash collection while others provided a comprehensive list of cost categories. Cash payments included, for example, transport, committee expenses, airtime (mobile telephones being used to report faults), petty cash and other office expenses, projects and maintenance. Given the level of aggregation in accounts and lack of detailed categorization of costs, it was not possible to categorically test the standard cost categories in the PBTC: overhead (15% sub-total operating costs), contingency (5% sub-total operating costs) and maintenance charge (2.5% of cost of bulk water). From an analytical review of accounts, there was little evidence they were unreasonable.

Anomalies recognized in accounts included unanticipated honoraria paid by four WUAs. These represented amounts paid to traditional leaders for their help on Executive Boards and conflicted with the voluntary basis of Board membership. In one WUA, the honorarium amounted to 11% of the cash collected and was nearly ten times the average monthly salary of water sellers. These honoraria payments detracted from margins intended for operations and maintenance and contravene the mandate for WUA establishment detailed in the Water Resources Act 2013.

Payments made in arrears were also visible across five WUAs. They included both payment of debts to BWB which had been inherited by WUAs when taking over kiosks and delays in payment such as salaries. Payment made in arrears by WUA since establishment and current arrears outstanding to BWB are documented in Table 4. We were surprised to learn that, on establishment, the WUAs inherit the outstanding debts at the kiosks from previous 'managers'. This seems counter-intuitive to the break-even foundation on which the PBTC is based. Essentially, historical debt is repaid from any surplus of actual returns against costs. Given the sensitive nature of this practice, no information was made available on the level of inherited debt on WUA establishment, and it was not possible to disaggregate payments. It was noted by those concerned that amounts outstanding to employees (wages) were paid first and the inherited debt was only ever paid from cash surplus. Five WUAs were struggling to alleviate arrears, due to their minimal, often negative, margins and their representatives expressed concern regarding this.

**Table 4.** WUAs arrears at May 2013 and since establishment (est.).

| | Year WUA est. | Accounts Available from (Date) | Arrears on Formation (MK) | ArrearsPaid to Date (MK) | ArrearsDue May 2013 (MK) |
|---|---|---|---|---|---|
| Bangwe | 2012 | 11/2012 | 5,607,506 | 0 | 5,607,506 |
| Michiru | 2010 | 07/2011 | Not available | Not available | 800,000 |
| Mitsidi-Sanjika | 2010 | 05/2011 | 9,735,446 | 5,855,570 | 3,879,876 |
| Mudi | 2011 | 12/2011 | 2,883,560 | 1,483,560 | 1,400,000 |
| Namiyango-Chigumula | 2010 | 01/2010 | 1,550,000 | 650,000 | 900,000 |
| Ndirande-Matope | 2009 | 04/2011 | 3,281,550 | 3,281,550 | 0 |
| Soche-Misesa | 2013 | 01/2013 | 527,600 | 527,600 | 0 |

Under the assumptions provided by the PBTC, the WUAs should breakeven at the 10 MK tariff set. Based on evidence drawn from the book-keeping systems of each WUA, including honoraria and inherited debt payments, a financial breakeven analysis was carried out to identify the tariff at which each WUA would have broken even based on their actual costs (including payments made to inherited arrears as these could not be disaggregated). This is detailed in Table 5, highlighting that, for three loss-making WUAs, a tariff of 12.35 MK, 14.62 MK and 15.03 MK would have been required to break even. These findings were shared with BWB and we expressed concern at the degree to which actual costs varied from standard with respect to unusual items such as honoraria. We recommended further training to help with appropriate cost allocation and accurate record keeping. As noted previously, payment of honoraria contravenes the mandate for WUA establishment detailed in the Water Resources Act 2013 and, on this specific point, we recommend further reforms are needed to deal with this (see Section 4, Discussion).

**Table 5.** Breakeven tariff calculation by WUAs.

| Bangwe | Michiru | Mitsidi-Sanjika | Mudi | Namiyano-Chigumula | Ndirande-Matope | Soche-Misesa |
|---|---|---|---|---|---|---|
| **Total Operating Costs (MK):** | | | | | | |
| 7,931,872 | 31,876,646 | 25,103,292 | 16,566,843 | 11,119,896 | 29,160,188 | 15,303,125 |
| **Volume of water sold per year (as number of 20 L buckets):** | | | | | | |
| 542,400 | 2,581,300 | 2,820,700 | 1,688,150 | 1,338,100 | 3,593,450 | 1,018,100 |
| **Estimated breakeven tariff (as MK per bucket):** | | | | | | |
| 14.62 | 12.35 | 8.9 | 9.81 | 8.31 | 8.11 | 15.03 |

All WUAs Secretariats reported contamination and interrupted supply as a contribution to poor financial performance and the most significant concern influencing access to water. Frustration was expressed by WUAs that contaminated water (typically coloured water due to suspended sediment arising from supply interruption) was charged for in a BWB metered supply, but could not be sold, and interrupted supply contributed to arrears forming. We advised that this contaminated supply should be recorded separately as per water meter readings and detailed in financial records so adjustments could be made for these by BWB once contamination was verified.

Comparing annual volumes of water sold by WUAs against population, we find in Table 6 that the daily per capita water consumption from the kiosk is shockingly low, ranging from 4.67 L at best, down to 0.53 L, well below minimum World Health Organization guidelines on consumption. Given these shockingly low levels, it is unlikely that the amount of bulk water drawn at the kiosk reflects demand. If average household consumption should be 20 L per person per day, as contained within the PBTC, this highlights that only around a quarter of the local population could be using this supply, or more are using it, but to a lesser extent than advised. This could be a result of some of the local population having their own connections, finding alternative supplies, or potentially being excluded by the 10 MK cost. Considerable dependence is, therefore, presumed to remain on other water sources. It was evident that people collected rainwater for washing, walked to the next functioning water kiosk and/or turned to black market supplies or frequented rural borehole supplies with 'free' water. Black market supplies are from illegal water sellers who either tamper with meters at the kiosk, and, as a result, pay less to BWB, or set up illegal, non-household, temporary taps in the piped network, and pay nothing to BWB. Of most concern were the people who resorted to drinking from unimproved water sources, especially untreated surface waters that were vulnerable to (microbiological) life-threatening contamination.

**Table 6.** Consumption of water from kiosks by WUA.

| WUA | Annual Volume of Water Sold (L) | Daily Consumption (L) | Population (n) | Daily Consumption Per Capita(L) | Estimated Population Coverage [1] (n (%)) |
|---|---|---|---|---|---|
| Bangwe | 10,848,000 | 29,720 | 34,611 | 0.86 | 1486 (4%) |
| Michiru | 51,626,000 | 141,440 | 56,892 | 2.49 | 7072 (12%) |
| Mitsidi-Sanjika | 56,414,000 | 154,560 | 47,898 | 3.23 | 7728 (16%) |
| Mudi | 33,763,000 | 92,500 | 49,561 | 1.87 | 4625 (9%) |
| Namiyango-Chigumula | 26,762,000 | 73,320 | 59,808 | 1.23 | 3666 (6%) |
| Ndirande-Matope | 71,869,000 | 196,900 | 42,130 | 4.67 | 9845 (23%) |
| Soche-Misesa | 20,360,000 | 55,780 | 104,893 | 0.53 | 2789 (3%) |
| TOTAL | 271,642,000 | 744,220 | 409,080 | 1.83 | 37,211 (9%) |

[1] For an assumed daily consumption of 20 L per capita of volume sold.

Given that the tariff at the kiosk was more than the cost of domestic water, it was not surprising to find evidence of water sales from unlicensed individual domestic supplies. Of more concern was the illegal sale of water from legal standpipes tapped into the

network pipes, adding to disruption. This was noted as a long-term problem influencing network supply and has been recognized in the literature on water supply in Malawi (see, for example, Cammack [33]). BWB took some responsibility for supply contaminations and interruptions, noting that the origin of the supply and its maintenance were being improved under a project funded by the European Investment Bank. Theft of cash was also evident, with inspectors carrying accumulations of cash and water-sellers being targeted. As a result, WUAs had increased security, resulting in a further cost to the WUA for a security guard. All such problems were recognized by BWB and WUAs as items for discussion by WUA Executive Boards.

*3.3. Negotiation of the Kiosk Tariff*

We found that tariff changes at the IBT level were subject to strict government approval and, to an extent, inflation by district and national government authorities. BWB had an influence on tariff setting to the extent that they influenced the cost inherent in the standard costing methods, could propose tariff changes and had fiscal responsibility for WUAs. In terms of the PTBC, when BWB had an IBT tariff change approved by government, it was a standard practice that meetings were convened through the KMU office to agree on a final tariff (IBT plus PBTC) at the kiosks with all WUAs operating in Blantyre. The meetings were seen as providing information and facilitating the negotiation and establishment of a uniform PBTC tariff. It was confirmed by NGO representatives that these meetings provided information regarding the block tariff change and negotiations on the final tariff to be charged by WUAs at all kiosks in Blantyre.

Further, debate between BWB and WUA representatives in formal WUA committee meetings revealed some noted 'negotiation' of standard costs, such as salary contributions, at kiosks, which ultimately would influence the PBTC and the kiosk tariff. In turn, change to the standard costing model reflects updates in standard costs, where salaries were uplifted and new standardized roles introduced. In this way, the WUA running costs influenced the standard PBTC model and ultimate kiosk tariff. Our evidence is in contrast to findings by Pihljak et al. [16], who found no formal evidence to suggest negotiations of local kiosk tariffs in Lilongwe.

*3.4. Developments over Time*

In 2013, BWB established a dedicated role of Kiosk Manager to "help sort out WUA problems". BWB's Kiosk Manager regularly visited all kiosks and became familiar with the local practices, problems and performance of WUA employees. They made BWB interest in WUAs visible and, given their knowledge of day-to-day activities, had the potential to inform tariff setting. At the same time, the NGO Water for People also increased their capacity to help BWB and WUAs by developing a role for a Training and Capacity Building Specialist to "supervise, inspire and motivate" WUAs.

Between 2013 and the end of 2016, no evidence was found of significant changes in the tariff's IBT or standard cost calculation (PBTC) beyond inflationary adjustments. Evidence was visible in Stakeholder Minutes and WUA Board Minutes of some agreement between the BWB Kiosk Manager and WUA Secretariats on how to prioritize costs and "chip away at arrears". However, there was evidence the WUA secretariats were frustrated by their inability to influence water supply caused by network problems. Engagement between the BWB Kiosk Manager and WUAs helped to increase BWB's knowledge of local conditions and associated financial position, which, in turn, supported informed decision-making across WUAs and, ideally, reduced costs and the tariff. In a WUA General Assembly Meeting in 2015, District Councilors attested that the Kiosk Manager, in particular, was "quite effective" and "a lot of issues had been turned around", but these issues were not detailed further.

Only one reference was found to the kiosk tariff in the Minutes, when a water seller asked for "assurance about the tariff being charged". The response was relatively dismissive, offering "reassurance to the seller"; no details were provided. The tariff appears

to be taken as a given and WUA meetings focus on discussing operations and reporting problems, with theft being a regular feature. While the Government Guidelines are clear, various stakeholders expressed concerns that community participation in WUAs was limited, and often populated by individuals driven by self-interest. Concerns were regularly raised with the General Assembly and WUA Executive Boards that, for example, "proper procedures of recruitment of staff are not followed", there had been "improper recruitment of WUA staff", and "political people are still being recruited ending up politicizing the Association". It is recognized by a representative of NGO Water for People that this later point on politicizing the WUA could be a special case but is thought not to be a general problem. Beyond WUAs, community interests were rarely represented in meetings. We found minimal reference to international NGOs, such as Water for People, nor the local, donor-funded, consumer association CAMA. For example, Minutes from a meeting concerning the need for new WUAs included CAMA stressing "that time for affiliating the kiosks to certain sections of the society was unfair and that it is important that all people in the community regardless of any social affiliation get a fair share in the management of kiosks in the area". This raises the questions of what they mean by fairness and how they will execute this. There was no discussion of what constitutes fairness. It was noted by Water for People representatives that their participation in such meetings was limited to avoid dependencies emerging between themselves and the WUAs.

At completion of the MDG era in 2015, the proportion of Malawi's population with sustainable access to an improved water source was estimated at 86.2%, up from 47% in 2000, with a rapid increase taking place in the last few years of the MDGs [2,48,49]. It was not possible to isolate specific increases in capacity for peri-urban Blantyre, but in new WUAs at Ndirande-Kachere and Ndirande-Malabada, increased capacity was evident through new water points at kiosks. In terms of network improvements, there was evidence of WUAs harnessing mobile phone technology to report problems and seek early solutions.

Providing new evidence, an NGO representative still working in the field attested that WUAs had started investing a lot in kiosk construction; for example, they cited Nkolokoti-Kachere as having built kiosks and water tanks to improve further service delivery using their own money, surplus gains from water sales, to pay for these investments. Further, they emphasized, "Nkolokoti WUA has built over 11 kiosks using their own resource and it had become standard practice in all WUAs that annually if they make some gains they invest into kiosks construction". To date, negotiation of the final kiosk tariff remains common practice facilitated by the KMU. Inheritance of old debt at the kiosks remains the main point of contention between BWB and WUAs.

## 4. Discussion: The Long-Term Sustainability of Kiosk Costs

Supplying water at kiosks through WUAs was intended to ensure equitable access to water and support payment of network costs given a history of local control/capture by political parties leading to non-payment. Our research in Blantyre Malawi between 2012 and 2016 found evidence that BWB had fiscal responsibility for tariff setting, subject to central government approval, and this included the incorporation of WUA costs. WUAs were responsible for financial record-keeping based on BWB guidance and a system of standard costing. BWB representatives supported training of WUA employees engaged in bookkeeping (by NGOs) and audited records in an attempt to exert control over practice and gain assurance of the accuracy of procedures. In turn, WUA record-keeping varied in standard and quality, with costs categories being added at WUA's discretion and there was evidence that money was sometimes misappropriated by WUA employees. Meter readings were matched with accounts to control for this, and it is noteworthy that any culprit found misappropriating money would have pay deducted from their monthly salary. Overall, no evidence was found of paying water users being denied water if the supply was functioning and uncontaminated; in this respect, the WUAs had achieved the mandate set out in the Malawi Water Resources Act 2013.

BWB prepared the kiosk tariff using an IBT (set centrally) and PBTC, conflating problems inherent in each. While the IBT applied to the kiosk by BWB (in May 2013) is 2.34 MK for delivery, a process of adding standard kiosk management costs for providing the connection takes the tariff charged to water users at the kiosk to 10MK without any test for or consideration of affordability. Particularly problematic is the credibility of the IBT, which is undermined when the kiosk tariff becomes higher than for domestic supply, which goes against IBT design principle, noted earlier, and it is evident that domestic supplies can be sold on more cheaply than comparable prices at kiosks. The level of the kiosk tariff should not be ruled out as a factor contributing to the illegal tapping of water points into the network pipes and cases of people accessing unsafe contaminated surface water sources.

While in Lilongwe, Pihljak et al. [16] found no formal evidence to suggest negotiations of local kiosk tariffs, we found contrasting evidence in Blantyre. We found that a different development and approval process occurred for each of the IBT and PBTC elements of the kiosk tariff. The IBT was set by BWB and subject to strict government approval. The PBTC was based on standard costs at the kiosk also set by BWB, who had fiscal authority for WUAs, and WUAs could comment on standard cost-setting through a formal structure of meetings built into their governance by the central Government. However, when BWB wanted to change the PBTC and, ultimately, the tariff at the kiosk, they did this through a meeting held with a collective of all WUAs through the KMU in Blantyre. This meeting served the purpose of information giving and tariff negotiation. Thus, the WUAs in Blantyre were arguably able to influence the tariff at the kiosk.

Establishment of WUAs by the government was intended to induce controlled community participation in kiosk management within a previously failing politicized system and to formalize accountability relationships and mechanisms. In Blantyre, the established WUA became the accounting entity for kiosks but hold an inter-mediator's role between BWB and the community. The costs underpinning tariff-setting and the WUA accounts used to justify this have a critical influence on the tariff's ability to achieve the goal of improving access to clean water. The giving and receipt of accounts, in terms of financial bookkeeping and the context within which they are created, is critical to negotiations of the tariff between BWB and the WUAs.

There is an argument for making WUAs more independent by giving them fiscal responsibility for the PBTC and making them responsible for kiosk tariff setting and running costs. Covering their own costs may give WUAs a greater desire to control costs, increase community participation and local accountability, and deter the potential misappropriation of money and water. However, this could lead to increased vulnerability to local political influence, as observed in the past. This may already be seen creeping in through the misappropriation of money for honoraria and, on this basis, we would advise against greater independence for WUAs. Instead, we propose that it would be more useful to invest in further training in bookkeeping and decision-making, in an effort to control costs. A private operator model of performance targets (including customer satisfaction) could also help here. Equally, we would deter BWB, increasing control over WUA activities (e.g., taking more in-house), as this could be seen as contravening the government's principle of community participation embedded in setting up WUAs. With this in mind, we favour the status quo in terms of fiscal responsibility and WUA discretion but advise that more investment is needed in training WUA employees on cost controls and its impact on tariffs.

Given that BWB had outstanding historical kiosk debts before the WUAs were established, responsibility for payment of this debt was attributed to WUAs, with repayment expected out of any surplus of tariff income over costs incurred. This was not factored into PBTC tariff setting. However, it has become standard practice for WUAs to pay the historic debts owed to BWB from any surplus while, for WUAs with a short fall between tariff income and costs, debts have become compounded. The inheritance of historical debt attached to kiosks by WUAs on formation has been recognised as a considerable point of

contention between BWB and WUA employees, and provided a disincentive to running WUA operations more efficiently and recovering more costs. Although strongly resisted by BWB, writing off historic debt and starting with a clean sheet is strongly advised when a new WUA is formed. Starting from this basis provides a much clearer insight into cost control and, in turn, arguably the ability to start to reduce the standard kiosk tariff which operates across the network as a whole. This is a condition which should be transparent and incorporated into any future legislation governing the not-for-profit nature of WUAs. Returning to the assumptions inherent within the first part of the tariff, the IBT that a minimal tariff is needed at the kiosk, we recommend that the IBT calculation is extended to integrate the kiosk costs. The costs will then enjoy the 'subsidy' built within the IBT and become more sustainable in the long term. In turn, the viability of WUAs managing the kiosk will also be sustained.

Despite WUAs establishment, the kiosk tariff was essentially imposed on community water users who, if afforded the opportunity for greater dialogue on its setting, may hold very different perspectives on WUA operations. Creating participation in water management through the employment of individuals certainly provides an opportunity for the alleviation of poverty for those employed, but it reinforces the hierarchy between the employed and unemployed, which could undermine a sense of community and community action. However, the alternative, no employment to keep things equal, would not be desirable. Employment arguably makes water tariffs more affordable at the kiosk, but this does not always help those who do not benefit from this employment. By virtue of extended family structure, there is arguably a cascade of benefits, theoretically speaking. Further channels are encouraged to engage the local community in decision-making. The payment of honoraria to community leaders goes against the operating principles of WUAs, which require volunteers on the board. However, as a representation of community involvement, we recommend the consideration of a mandate for a 'worthwhile but not excessive' fixed payment to Executive Director's from the community on the WUA Board. The associated increase in costs resulting from a fixed payment could be countered by the greater efficiency of WUA management and improved water-point functionality.

The central role and function of WUA's financial accounts, complemented by local knowledge of everyday life, are critical to evaluating costs underlying tariff setting and access to clean water. However, evidence from formal WUA meetings illustrates the limited expression of community views and a lack of community voice. Arguably the most destructive element in the BWB–WUAs relationship was contention over inherited arrears. This appeared time and time again in the WUA Minutes, and is a point of disagreement which BWB should reconsider. It is not within BWB's interest for WUAs not to succeed, and inherited debt is outside WUAs control and arguably due to the ineffectiveness of BWB in handling debtors. There is no statutory guidance on this legacy debt, and we propose consideration in the revised Water Resources Policy and the upcoming review of the Water Works Act (Malawi). Further, a trusted partnership with a greater voice given to the community should be encouraged, to enable greater insight into the affordability of tariffs and ensure sustainable access to water.

## 5. Conclusions and Recommendations

Overall, the premise of public water delivery is to charge the poor an affordable tariff to cover the cost-of-service delivery. Inability to pay could put pressure on already vulnerable people to seek alternative sources, which may not be suitable, particularly for drinking. Our empirical contribution is based on action research in the field in the pursuit of sustainability. It involves a longitudinal study of kiosk tariffs operating within the impoverished communities of peri-urban Blantyre, Malawi, from 2012 to 2016, encompassing the MDGs commitment period ending in 2015 and a move to the SDGs. We provide evidence of calculative practice in tariff setting that encompasses economic and accounting by BWB, who have fiscal responsibility for the water supply and WUAs mandated to operate networked kiosks.

We find that a standard costing system operates and the tariff increment for WUA kiosk operation is set at a breakeven point of tariffs collected and costs. This is in line with government guidelines, as a WUA should operate on a not-for-profit basis. However, considerable disparity is revealed between the standard costs and the actual costs incurred by different WUAs, resulting in surplus and deficits. We find BWB encourages a surplus in order to pay off historic debt recorded at the kiosk and inherited by WUAs on establishment.

We recommend that historic debt is written off and should not be a cost for new WUAs, as this acts as a disincentive to manage costs appropriately, for both the WUAs and BWB. There is no statutory guidance on legacy debt, and we propose that future reforms to the Water Works Act (Malawi) and updates to the National Water Policy should take this into account. We note a primary discrepancy between the costs incurred by WUAs and government policy, in the form of honoraria payments being made to local leaders by WUAs. We recommend further reforms to deal with this. Our main concern is that kiosk tariffs are exceeding residential water tariffs, and reference is made to residential sales and illegal tapping of the water network which could be fueled by this. We further recommend that water boards work with WUAs and provide additional training for WUA employees to help re-evaluate running costs and, ultimately, reduce tariffs to avoid people turning to unsafe alternative sources. Of more significance, we highlight the need for BWB to include kiosk costs into their integrated block tariff calculations so they are covered accordingly, ensuring that the tariffs are sustainable.

While community participation in water supply through WUAs has arguably improved access to water, it has led to considerably higher tariffs at the kiosk. The costs of a sustainable water supply delivered under government mandate by WUAs at network kiosks need to be fully integrated into tariff decision-making for the network as a whole (in this case, taken within the IBT, which forms the current theoretical underpinning for the main network tariffs). This will ensure that kiosk tariffs remain below those for a domestic supply and make costs more accessible to people living in poverty, in compliance with SDG-6. Future research into water kiosk tariffs and WUA accounting needs to move beyond traditional standard costing methods to more dialogic accounts of the value of water in everyday life, as provided by water users, thus engaging communities in accounting for social impact and sustainability.

**Author Contributions:** Conceptualization, A.B.C., S.M.P.F., M.O.R., R.M.K.; methodology, S.M.P.F. and A.B.C.; formal analysis, A.B.C., S.M.P.F. and M.O.R.; investigation, S.M.P.F., J.P.T. and A.B.C.; resources, R.M.K.; data curation, A.B.C., M.O.R. and S.M.P.F.; writing—original draft preparation, A.B.C. and M.O.R.; writing—review and editing, A.B.C., M.O.R., S.M.P.F., R.M.K., J.P.T., M.N. and J.M.; supervision, A.B.C. and R.M.K.; project administration, R.M.K.; funding acquisition, R.M.K. All authors have read and agreed to the published version of the manuscript.

**Funding:** This research was funded by the Scottish Government Climate Justice Fund Water Futures Programme research grant HN-CJF-03 awarded to the University of Strathclyde (R.M.K.).

**Institutional Review Board Statement:** The study was conducted in accordance with approved University of Strathclyde Ethics Committee guidance.

**Informed Consent Statement:** Informed consent was obtained from all subjects involved in the study.

**Data Availability Statement:** Not applicable.

**Acknowledgments:** The authors would like to acknowledge support from the Scottish Government Climate Justice Fund Water Futures Programme, University of Strathclyde, Water for People and BASEflow. We acknowledge that the research includes work carried out by Sergio M.P. Fernández as part of his MSc (Sustainability and Environmental Studies) dissertation 'Water Management and Water Tariff Structure in Peri-Urban Blantyre' and Jonathan P. Truslove both while at the University of Strathclyde. We note the findings, interpretations and conclusions expressed in this work do not necessarily reflect the views of their current employers. We would also like to thank funders from the EIB, EU and Tilitonse Fund for their local support of water resources in Blantyre where the CJF Programme was running.

**Conflicts of Interest:** The authors declare no conflict of interest.

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
