# Peer review of "The Cost of a Sustainable Water Supply at Network Kiosks in Peri-Urban Blantyre, Malawi"

_sustainability, doi:10.3390/su13094685_

Round 1

Reviewer 1 Report

The topic is really interesting and the scarcity of scientific articles on such an interesting subject as the selling price of water by kiosks highlights the importance of this aspect.

The study has also social aspect not negligible considering the role of drinking water for life, work and primary use. Fiscal aspects carry out a role into the sale price definition and in these countries a careful evaluation must be involved for these services.

The paper is well written but for some aspects could be improved.

For instance, about Tariff calculation prices are express into Malawian Kwacha however is difficult for international researchers to appreciate and understanding effective prices using a local money. I suggest to express that in Dollars. 

Author Response

Thank you for your valuable review, it is much appreciated. Please see attached for our detailed response to your comments on a point by point basis.

Reviewer 2 Report

This study is on evaluating Kiosk tariffs for water supply. The study is interesting and worth publication. 

Author Response

Thank you for your very supportive review. It is much appreciated. Please see attached our specific response on the point which you made.

Reviewer 3 Report

Dear authors,

In your introduction, you do not discuss a number of important issues such as the research gap of your study, the study’s research question and objectives, and the contribution of this study. These issues should be discussed thoroughly before you move to the literature review which usually includes the theoretical background, hypotheses or propositions’ building, and the study’s conceptual model.

The increase in water demand and the analysis of its cost is a global problem that needs to be studied in virtually all countries, I suggest you include this point of view in your study supported by recent scientific literature, e.g.

Pérez, D. M. G., Martín, J. M. M., Martínez, J. M. G., & Sáez-Fernández, F. J. (2020). An Analysis of the Cost of Water Supply Linked to the Tourism Industry. An Application to the Case of the Island of Ibiza in Spain. Water, 12(7), 2006.

I think that the key objective of your paper is not very clear, so in the introduction, you can try to answer some points of interest. 1. An effective introduction answers three sets of questions: a. Who cares? What is the topic or research question, and why is it interesting and important in theory and practice? b. What do we know, what don't we know, and so what? What major, unaddressed puzzle, controversy, or paradox does this study address, and why does it need to be addressed? c. What will we learn? How does your study fundamentally change, challenge, or advance scholars' understanding?

Also, It would be a point of improvement to include a map with the studied areas of Malawi, their location, and the distance between them.

Theoretical framework

In your study you do not try to develop a theoretical background and your study’s conceptual model is not discussed at all. You do not have a theoretical framework which to be related to help reduce the kiosk tariff as you are discussing financial control which associates with the reduction of WUAs costs and the kiosk tariffs WUAs. need more training in financial record keeping and cost management. You need to search the very good literature on developing new procedures of financial record keeping and cost management that you can find a suitable model related to your findings and then develop and discuss the propositions of your research model.

Methodology

The methods section is not clear. More detail is required regarding all the key aspects of the methods. For example, more detail is required regarding all the kiosk tariffs WUAs, and how exactly accounting was implemented in each zone. What was the role of the person interviewed in this tariff implementation? The data analysis is not clear.

I recommend including as a note the Forex rate of kwacha against US dollar.

Results and discussion: It is recommended to indicate the program that is used for financial analyses.

Conclusion: As your findings are interesting ones, you can suggest a conceptual model based on them. This conceptual model can be tested in future research. However, you can at least discuss and develop some propositions based on the relationships of your conceptual model. In your Conclusion, you do not discuss theoretical and managerial implications, though you discuss limitations and future research.

Finally, it could be interesting to highlight the practical significance for organizational members of this study and make the reference to policy prescriptions that derive from this analysis, as well as the implications for future research. I will encourage the authors to expand the agenda for future research.

Author Response

Thank you for your valuable review. Your comments are very helpful and much appreciated. Please see attached our detailed response on the specific points which you made.

Round 2

Reviewer 3 Report

Many thanks to the authors for the new manuscript, which reflects the suggestions made for the improvement of the work presented.
Congratulations.
Kind regards.